# Breakdown or Personality Disorder? Psychiatric Characterization and Developmental Pathways Towards Young Adulthood in Adolescents with Pathological Personality Structure

**DOI:** 10.3390/brainsci14111115

**Published:** 2024-11-01

**Authors:** Chiara Rogantini, Marika Orlandi, Arianna Vecchio, Diletta Cristina Pratile, Raffaella Fiamma Cabini, Elena Ballante, Valentina De Giorgis, Renato Borgatti, Martina Maria Mensi

**Affiliations:** 1Department of Brain and Behavioral Sciences, University of Pavia, 27100 Pavia, Italy; 2Child Neurology and Psychiatry Unit, IRCCS Mondino Foundation, 27100 Pavia, Italy; 3National Institute for Nuclear Physics (INFN) Section of Pavia, 27100 Pavia, Italy; 4Department of Political and Social Sciences, University of Pavia, 27100 Pavia, Italy

**Keywords:** personality disorders, adolescence, high risk, transition, follow-up, prevention

## Abstract

**Background/Objectives**: Research is lacking about the development of personality disorders (PDs) from adolescence to early adulthood. This study aimed to characterize the profile of high-risk adolescents compared to adolescents with full-blown PDs and adolescents with other psychiatric disorders and to identify clinical markers that constitute a risk profile. **Methods**: We evaluated 99 adolescents (12–17 years old) through DSM-5-based semi-structured interviews, questionnaires, cognitive tasks, and scales regarding functioning and disorder severity. We divided patients into three groups: 22 adolescents with full-blown personality disorder (FBPD), 57 adolescents at high risk for personality disorders (HPD), and 20 adolescents with other DSM-5 psychiatric disorders (OTH). At follow-up, 56 patients completed the assessment. Possible developmental trajectories of FBPD and HPD patients were remission (total or partial) of PD-related symptoms, stability of symptoms, or transition from HPD to FBPD. **Results**: FBPD adolescents had more impaired family backgrounds. At baseline, the HPD group was mainly composed of female patients, younger than the FBPD ones. Externalizing symptoms may represent prodromal symptoms of FBPD. High-risk patients who made the transition were younger than those who did not, and their mothers reported higher internalizing symptoms at baseline. None of the FBPD patients remitted. **Conclusions**: These findings support the hypothesis of a PD “at-risk mental state” and the importance of the implementation of PD early detection and treatment in adolescents, regardless of patient age.

## 1. Introduction

Personality disorders (PDs) are common and clinically relevant psychiatric disorders among adolescents (12–18 years) and youth (15–25 years) [1]. Some young people can present before the age of 18 with severe and long-lasting symptoms that are sometimes refractory to treatment. Recent research indicates that PDs comprise both stable and dynamic processes over time. While many of the acute symptoms of PDs remit during the first few years, underlying personality traits may persist for extended periods. They may not be fully addressed by current treatments [2]. Furthermore, long-term functional recovery is often challenging. Adolescents face barriers to primary and secondary care access, delays in receiving appropriate and evidence-based treatments, and gaps between child and adult mental health services [3]. This emphasizes the importance of making early interventions available to enable patients to resume a healthier trajectory toward social and occupational functioning early on, which can be challenging for older patients [4]. As such, it is crucial to search for clinical indicators of psychopathological risk in adolescent patients who present with a diagnosis of unspecified PD or subthreshold PD symptoms to activate the most appropriate treatment strategies promptly. Recent evidence [5] states that subthreshold PDs, particularly those related to borderline personality disorder (BPD), are also associated with more significant psychosocial morbidity.

Despite the international consensus that PDs can reliably and validly be diagnosed in young people and recent evidence showing that even subthreshold PDs are associated with poor outcomes, some issues remain vastly unexplored. In the framework of a preventive staging model, proposed definitions to establish theoretical and practical thresholds to assess PD severity and developmental pathways are inconsistent between authors [6]. Furthermore, as mental disorders frequently present with evolving symptom mixtures in young patients, prodromal or unspecified personality symptoms are not usually captured by the categorical DSM-5 PD diagnoses [6]. Moreover, the criteria for distinguishing patients with and without a PD are arbitrary, and there is no strict distinction between these categories [7]. When symptoms are subthreshold, early application of a specific diagnosis might be inappropriate, and clinicians frequently avoid diagnosing PDs in children and adolescents, partly because of concerns about stigma [8,9]. Misleading terms or substitute diagnoses such as “PD features”, “atypical personality”, and “personality pathology” are often applied, leading to a delay in the diagnosis process of PDs and denying individuals the opportunity to make informed and evidence-based treatment decisions.

To the extent of our knowledge, there are no studies that characterize, in terms of psychological functioning, adolescents affected by a full-blown PD, from adolescents with unspecified PDs.

## 2. Materials and Methods

### 2.1. Aims

The present study had three aims:To characterize, from a psychiatric perspective, the profile of adolescents at high risk for PDs, compared with adolescents with a full-blown PD and with other psychiatric disorders.To evaluate the developmental trajectories of high-risk patients by assessing the transition to full-blown PD from baseline to follow-up.To identify clinical markers, features, and variables at baseline that may constitute a risk profile for progressing to a full-blown PD and represent specific targets of prevention therapeutic programs.

### 2.2. Study Design

This clinical register-based prospective cohort study was conducted according to the Strengthening the Reporting of Observational Studies in Epidemiology (STROBE) Statement (Appendix A). The authors assert that all procedures contributing to this work comply with the ethical standards of the relevant national and institutional committees on human experimentation and with the Helsinki Declaration of 1975, as revised in 2008. All procedures involving human subjects/patients were approved by the Ethics Committee of Policlinico San Matteo in Pavia, Italy (P-20170016006). The pseudonymized data are available in the Zenodo repository.

### 2.3. Study Population

Ninety-nine help-seeking adolescent inpatients and outpatients aged 12–17, admitted to the Child and Adolescent Neuropsychiatry unit of IRCCS Mondino Foundation (Pavia, Lombardy, Italy) were recruited between October 2012 and September 2021. This tertiary-level healthcare facility provides a specialized, highly technical level of healthcare that includes diagnosis and treatment of psychiatric disease and only assesses adolescents referred by secondary mental health services located in the local community or primary care units. Outpatients accessed the facility on a weekly basis to complete psychiatric assessment.

Exclusion criteria were as follows: (I) history of head trauma or any other underlying medical or neurological condition, (II) addiction to illicit substances or illicit substance-induced mental disorders, (III) ascertained intellectual disability (IQ ≤ 70). Participants and their legal guardians provided written consent and were free to withdraw at any moment. Figure 1 shows the final study population.

### 2.4. Study Measures

#### 2.4.1. Baseline Variables

All participants underwent a comprehensive assessment upon recruitment into the study, including the collection of sociodemographic data, exposure to psychopharmacological or psychotherapeutic treatment, and family and personal history of any DSM-5 psychiatric disorder [10].

We performed an extensive diagnostic assessment comprising clinician-rated semi-structured interviews, including the Kiddie Schedule for Affective Disorder and Schizophrenia Present and Lifetime DSM-5 (K-SADS-PL) [11,12], which assesses the presence of diagnostic comorbidity and the Structured Clinical Interview for the DSM-5 Personality Disorder (SCID-5-PD) [13]. This semi-structured interview provides categorical diagnoses (present or absent) of PDs based on the DSM criteria. For each disorder, the clinician is asked to indicate whether the threshold for categorical diagnosis has been met (e.g., at least 4 out of 7 criteria for avoidant personality disorder). If the categorical threshold for a particular disorder has not been met, the clinician may still indicate the presence of significant subthreshold clinical features. The interview concludes with the possibility of diagnosing an “unspecified personality disorder” when features of several PDs are present that do not meet the criteria necessary to diagnose any specific disorder but still cause significant impairment in functioning. At baseline, PDs were assessed using the Structured Clinical Interview for DSM-IV Axis II Personality Disorders (SCID-II Structured Clinical Interview for DSM-IV Axis II Disorders) [14] and later, after 2017, with the Italian version of the updated Structured Clinical Interview for DSM-5 (SCID-5-PD) [13]. All diagnoses were revised and updated according to DSM-5 criteria to harmonize data from different manuals. We proceeded the same way to harmonize categories of the different versions of the SCID.

Clinicians also assessed the overall severity of the disease by filling in the Clinical Global Impression-Severity (CGI-S) [15] and the patient’s general level of functioning using the Children’s Global Assessment Scale (CGAS) [16]. In addition, the IQ was assessed by the Wechsler Intelligence Scale of Children (WISC-IV) or the Wechsler Adult Intelligence Scale (WAIS-IV), depending on age [17,18]. Adolescents self-reported emotional and behavioral problems filing in the well-validated Italian version of the Youth Self-Report (YSR) [19], while mothers and fathers filled in separately the Child Behavior Check-List (CBCL) [19], which quantifies internalizing (INT), externalizing (EXT), and total (TOT) behavioral problems. Each statement is rated on a Likert scale as follows: 0 (not true), 1 (somewhat or sometimes true), or 2 (very true or often true). The continuous T scores can be categorized as clinical risk (above 64), borderline risk (between 60 and 64), and no risk (below 60).

At baseline, after diagnostic assessment, according to the SCID-5-PD and the K-SADS-PL results and the DSM-5 diagnostic criteria, participants were divided into three groups: (I) adolescents with a full-blown personality disorder (FBPD) (*n* = 22); (II) adolescents at high risk for personality disorders (HPD) (*n* = 57); and (III) adolescents with other DSM-5 psychiatric disorders (OTH) (*n* = 20).

Specifically, patients included in the FBPD group met the DSM-5 criteria for a PD diagnosis. On the contrary, the HPD group consisted of patients who, according to the SCID-5-PD results and the DSM-5 diagnostic criteria, presented with an unspecified personality disorder or subthreshold personality symptoms; patients with personality disorders that do not allow the diagnosis of specific PD due to their young age; and patients with misleading diagnoses.

#### 2.4.2. Follow-Up

Follow-up of FBPD and HPD patients lasted 1 to 10 years from baseline assessment (mean of 5.5 years). Participants were asked to participate in the follow-up assessment after 1 year from baseline; if they declined to participate, we tried to contact them every year. Those who always declined to participate in the follow-up assessments, were considered dropouts. Follow-up assessments were conducted remotely using home interviews with parents and patients, except in the cases of total or partial unreliability, and with the psychiatrist or child and adolescent psychiatrist where possible.

A clinical interview was conducted with parents and, consequently, with the patient, including more specific questions regarding mood, the presence of alterations of thought and perception, and a clinical interview to investigate the level of functioning (CGI, CGAS). All patients then underwent the SCID-5 PD. Based on the information gathered, it was possible to identify the following developmental trajectories of FBPD and HPD patients: (I) remission (total or partial) of PD-related symptoms previously detected at baseline; (II) stability of PD-related symptoms detected at baseline; (III) transition from HPD to FBPD.

### 2.5. Data Analysis

Descriptive analyses for discrete variables included absolute frequencies for each class of the variable and relative frequencies (proportions of the total number of observations). The median, first and third quartiles, mean, and standard deviation (SD) were calculated for continuous variables. Statistical comparisons among the three groups supplemented descriptive analyses. The Kruskal–Wallis test was used for numerical variables and Chi-squared tests for discrete variables, supplemented by post hoc analysis (Dunn’s and Fisher’s tests, respectively). Bonferroni correction was applied to all post hoc analyses to reduce the probability of type I errors due to multiple testing. To compare the HPD group that made the transition with the HPD group that did not, we used linear regression models for continuous variables, logistic regression models for binary variables, and ordinal regression models for ordinal categorical variables. In each model, the influence of the groups was adjusted for age, gender, and follow-up time. The statistical significance level was *p* ≤ 0.05 for all the statistical tests. Transition to FBPD was described through the Kaplan–Meier failure function (1-survival), complemented by 95% CIs, calculated through the standard error of the cumulative hazard, i.e., log (survival). Censoring was defined when patients did not transition at the last follow-up clinical observation. All statistical analyses were conducted in R 4.0.2, RStudio [20], and figures were produced using the package ggplot2 [21].

## 3. Results

### 3.1. Cross-Sectional Between-Group Analysis at Baseline

Socio-demographic and family history of psychiatric disorders results relative to the total study population and the three subgroups are reported in Table 1.

Personal history of DSM-5 psychiatric disorders, baseline exposure to psychiatric treatments, self-administered questionnaires, and functioning results relative to the total study population and the three subgroups are reported in Table 2.

### 3.2. Cross-Sectional Analysis of Dropout vs. Non-Dropout Group

Of the 99 patients included at baseline, 43 were lost at follow-up. We retrospectively considered the patients who dropped out at follow-up (*n* = 43) and compared them to the patients who continued to participate in the study (*n* = 56) to assess whether they showed different profiles at baseline. Results showed no difference at baseline between the dropout and the non-dropout subjects, except for the PD diagnosis (see Appendix A). Indeed, all the patients in the non-dropout group belonged to the HPD (71.4%) or the FBPD (28.6%) groups, while all the OTH adolescents recruited at baseline (46.5%) dropped out, together with some HPD (39.5%) and FBPD (14%) patients. From baseline to follow-up, 27.3% of the FBPD patients and 29.8% of the HPD ones dropped out.

### 3.3. Analyses at Follow-Up

The final follow-up population comprised 56 subjects, 16 from the FBPD group and 40 from the HPD group. By comparing the PD diagnosis, according to the DSM-5 criteria, at baseline and follow-up, participants were divided into three groups: (I) adolescents who maintained the full-blown PD diagnosis from baseline to follow-up (SPD) (*n* = 16); (II) adolescents who transitioned from a high-risk status for PDs to a full-blown PD (TPD) (*n* = 13); and (III) adolescents who did not transition from a high-risk status for PDs to a full-blown PD (non-TPD) (*n* = 27). The non-TPD group included both participants who maintained a high risk for PDs from baseline to follow-up (*n* = 13) and participants who remitted (*n* = 14). Of the HPD patients who completed the follow-up, 32.5% developed a full-blown disorder (TPD), while 67.5% maintained their high risk or remitted (non-TPD).

Participants in the three groups were retrospectively compared at baseline. Since the follow-up interval was highly variable among adolescents in our clinical sample, results were corrected for follow-up time, age, and sex.

Results showed that, at baseline, TPD patients showed significant differences from the non-TPD ones in the socio-demographical domain and the self-report questionnaires and from the SPD ones in the self-report questionnaires. No significant differences were found among the three groups concerning the other variables considered.

#### Socio-Demographics and Self-Administered Questionnaires

The age of the TPD group was significantly lower than that of the non-TPD patients (see Table 3). The groups did not show significant differences relative to other socio-demographic features.

Concerning the reports of the CBCL questionnaire completed by the mothers, they were found to be significantly higher in the TPD group than in the non-TPD group relative to the scores in the internalizing problems (INT) domain (see Table 3). There were no statistically significant differences among the three groups in the father’s CBCL reports.

Regarding the YSR reports, the TPD group showed significantly higher INT score related to internalizing problems than the SPD group (see Table 3).

## 4. Discussion

The first aim of the present study was to characterize from a psychiatric perspective the profile of adolescents at high risk for PDs, compared with adolescents with a full-blown PD and with other psychiatric disorders.

First, among the group of patients with full-blown PD, a high proportion (86.36%) of adolescents met criteria for borderline PD: this aligns with evidence indicating that in adolescence, as for the adult age, BPD results as the most frequently diagnosed and studied PD [22,23]. Although the proportion of other PDs was scarce in our sample, we suggest, as proposed by Sharp and Wall [24], that our findings and conclusions may apply to personality pathology more generally, based on recent evidence demonstrating that borderline pathology represents core or shared features of personality pathology [25,26,27].

In addition, at baseline, high-risk patients exhibited a higher IQ than participants with a full-blown personality disorder. This result is consistent with studies emphasizing that a low IQ is a risk factor for developing a full-blown PD [28]. Low IQ has also been associated with a poorer prognosis: these patients seem more likely to come across poor functional outcomes, such as low educational qualifications, at 18 years of age [29].

Our study identified a psychiatric characterization of high-risk patients at baseline: the group was mainly composed of females of Italian nationality, primarily from families of separated or divorced parents. These patients differed from those who already received a diagnosis of a full-blown disorder in terms of a less impaired family condition: in particular, high-risk patients exhibited a lower frequency of familiarity with substance use disorder in first-degree relatives and a lower frequency of severe family problems (e.g., a parent living in prison, or a parent living in a community for severe drug addiction problems). This result was in line with the evidence that a severely impaired family background is a risk factor for the development of a full-blown PD [30,31]. Furthermore, high-risk patients were younger than full-blown patients at baseline, in agreement with recent models of PD clinical staging that propose the existence of a prodromal “at-risk mental state” preceding the development of full-blown symptoms [6].

Moreover, high-risk patients reported higher internalizing problems scores on the YSR questionnaires than full-blown patients. As suggested by some authors [24,30], it is likely that such patients, being at an earlier stage of personality disorder, exhibit more significant impairment in terms of associated internalizing symptoms.

More specifically, internalizing problems begin to emerge in pre-adolescence in the form of anxiety and depressive symptoms and, if untreated and in the context of predisposing biological vulnerabilities and stressful life events, provide a platform for the development of personality pathology in adolescence [24]. It is also possible to hypothesize that internalizing symptoms in the full-blown stage of PDs are still present but become more ego-syntonic due to the poor insight that often characterizes this stage [31] and is, therefore, not directly reported by patients as disturbing and problematic. In line with this theory, results of our follow-up analyses indicate that, at baseline, TPD adolescents showed significantly higher YSR INT scores than the SPD ones. This finding could reflect a higher insight of internalizing symptoms, perceived as ego-dystonic, in the high-risk participants who later developed a full-blown disorder compared to those who already presented with a PD.

Our study also unveiled that high-risk patients’ mothers, compared to those of patients with other psychiatric disorders, more frequently report externalizing problems in the CBCL self-report questionnaire. This appears to be in line with Sharp and colleagues’ model, which emphasizes that externalizing symptoms may also represent prodromal symptoms of subsequent full-blown PD.

High-risk patients did not differ significantly from patients with full-blown disorders at baseline in terms of overall level of functioning and disease severity. The prodromal stage is not associated with less impaired functioning. This finding aligns with recent evidence [5] that states that even subthreshold PDs are associated with more significant psychosocial morbidity. Thus, the level of impaired functioning does not help distinguish prodromal from full-blown disorders at baseline.

On the other hand, functioning was found to be more impaired in full-blown patients than in patients with other psychiatric disorders without personality alterations. This result confirms the urgent need for early intervention for adolescents suffering from personality psychopathology, as the level of related functional impairment is exceptionally high [32].

Our study disclosed that full-blown patients exhibited more frequently severe risky behaviors compared to patients without personality alterations. It could be suggested that the sample of full-blown patients was mainly compounded by patients who have borderline personality disorders, which are traditionally associated with risky behaviors, as well as with a greater tendency to unveil externalizing behaviors [33].

The second aim of the current study was to evaluate the developmental trajectories of high-risk patients by assessing their transition to full-blown PD from baseline to follow-up.

First, the patients who dropped out and those who adhered to the follow-up assessment were compared. A significant difference emerged in the PD diagnoses; indeed, in the FBPD and the HPD groups, approximately 1/3 of patients dropped out, while all the participants in the OTH group dropped out. This difference is possibly due to a lack of interest and personal involvement in the topic in the subjects who did not present a specific risk for PDs at baseline. Among the 56 participants who completed the follow-up assessment, all the FBPD patients maintained their PD diagnosis. This finding suggests that when a structured personality disorder presents at a young age, patients have a low probability of remitting later in time. Since most of our sample was constituted by borderline personality disorders, this result could appear to be in contrast with evidence indicating that BPD tends to become less evident or to remit with age [10]. However, our follow-up time was relatively short and did not allow concluding this sense.

Among the 40 high-risk patients, three trajectories were detected over time: (I) patients who went into complete remission of PD symptoms, (II) patients who remained clinically and diagnostically stable over time, and (III) patients who went on to complete PD. These findings align with high rates of remission and change reported for personality pathology, similarly in adults and adolescents [32]. This is particularly true for some types of PDs (e.g., BPD, as in our sample) that tend to become less evident or remit with age [10]. Therefore, many adolescents and young adults who revealed personality psychopathology may have a similar level of general functioning as people without a history of personality disorders [9]. Declines in symptom levels from adolescence through early adulthood are consistent with the hypothesis that many people “outgrow” PDs during the transition from adolescence to adulthood due to maturation and socialization [9].

Interestingly, 13 high-risk patients made the transition to full-blown PDs: this means that in our sample, approximately one out of three adolescents who, at baseline, presented with high risk for PDs had received a PD diagnosis at follow-up. Given the high percentage of transition to full-blown disorders, these findings are clinically relevant and highlight the need to implement early detection and intervention programs worldwide for the ”high-risk for PD” clinical condition, enabling diagnosis and treatment when an individual first meets DSM-5 criteria for the disorder, regardless of their age [34].

The third aim of the current study was to identify clinical markers, features, and variables at baseline that may constitute a risk profile for progressing to a full-blown PD and that might represent specific targets of prevention therapeutic programs. Our study documented that, among high-risk patients, the TPD ones were younger at baseline than the non-TPD ones. This result builds upon the finding that, at baseline, HPD patients were younger than the FBPD ones. Taken together, these results suggest that the younger the age of patients who show subthreshold PD symptoms, the higher the risk that they will develop a full-blown PD. This finding is highly relevant because it provides an evidence-based clinical feature to consider in clinical practice.

Moreover, the TPD patients’ mothers reported higher CBCL internalizing symptoms scores at baseline compared to the non-TPD ones. This finding suggests that, even among young adolescent patients, a high level of clinical impairment and symptomatic burden relative to internalizing aspects can be found [24], which, as discussed above, often characterizes the prodromal phase of the disorder. This result confirms that clinicians should maintain an open attitude toward early diagnosis, avoiding misleading terms, thereby eluding diagnostic delays.

Finally, there were no statistically significant differences at baseline concerning the main diagnostic instrument, the SCID-5 PD, between the TPD and non-TPD patients. Therefore, our results suggest that the SCID-5 PD is not helpful for clinicians in profiling patients at greater risk of transition than others.

## 5. Conclusions

Taken together, our findings indicate that adolescents, who, according to the SCID-5-PD results and the DSM-5 diagnostic criteria, present with an unspecified personality disorder or subthreshold personality symptoms, can be considered at high risk for developing PDs. Among high-risk adolescents, the younger ones, and those whose mothers report higher CBCL internalizing problems scores, may be at higher risk to develop a full-blown personality disorder later in time.

The results of the current study provide scientific advances in the field of adolescent personality psychopathology. To the extent of our knowledge, there are no longitudinal studies among adolescents with high risk for PDs and no studies evaluating the risk of transition to full-blown PD. To date, this is the most extensive real-world prospective cohort study with the most comprehensive clinical assessment and longest follow-up addressing the presentation and outcomes of adolescents meeting unspecified PD criteria, full-blown PD, and other psychiatric disorders, and it is the first to report the transition risk in this population. Furthermore, the current study is innovative since the sampling frame for our adolescent cohorts entirely consisted of adolescents with severe symptoms as clinical entrants of our tertiary-level healthcare facility rather than participants solicited via a research selection.

Among the study’s limitations, we only enrolled inpatients and outpatients from a tertiary-level facility, which is generally more severe. Concerning the study population, the sample size is relatively small, and we did not designate a control group; furthermore, we registered a high rate of drop out among our subjects. Another limitation could be the high variability in the follow-up time.

The current study’s findings confirm that clinical priorities should include early detection and intervention of adolescent PD and implementing diagnosis and treatment when individuals first meet DSM-5 criteria, regardless of age [34]. On the contrary, the younger the age that individuals present with a high risk for personality disorders, the higher the possibility that they develop a full-blown disorder. Prevention should, therefore, target high-risk individuals, representing the best starting point for creating a comprehensive prevention strategy. Our study supports recent evidence and clinical staging models that hypothesize the presence of an “at-risk mental state” for personality disorders, corresponding to a prodromal stage of illness before it meets the diagnostic criteria. These patients might present high impairment in terms of self-perceived internalizing symptoms and general psychiatric symptoms perceived by their mothers. These elements deserve to be considered valuable targets for intervention. As such, we suggest training mental health professionals in evidence-based early interventions to be prioritized and families actively involved as collaborators in prevention and early intervention [34].

## Figures and Tables

**Figure 1 brainsci-14-01115-f001:**
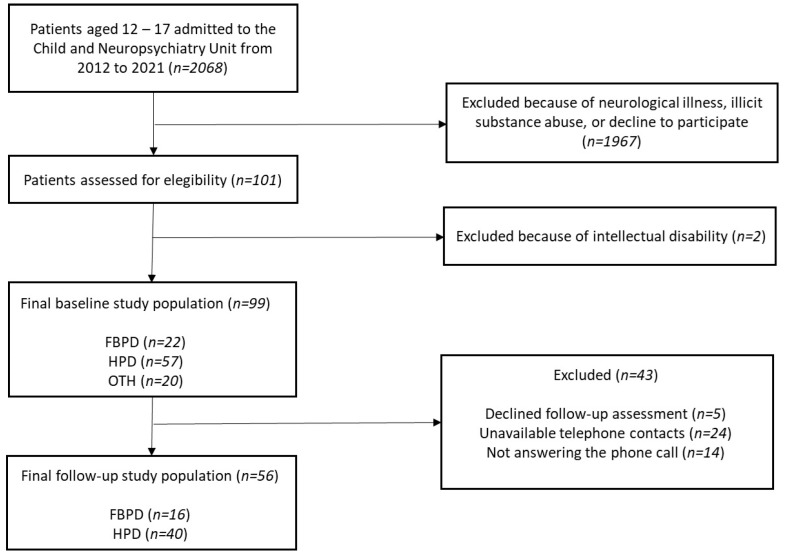
Flow chart of the study population. [Author’s own processing].

**Table 1 brainsci-14-01115-t001:** Socio-demographics and family history of psychiatric disorders in the total adolescent sample and the three subgroups [author’s own processing].

Characteristics	Total (*n* = 99)	FBPD (a) (*n* = 22)	HPD (b) (*n* = 57)	OTH (c) (*n* = 20)	*p* ^1^	Post Hoc
Socio-demographics						
Age, months, mean (SD)	189.17 (16.16)	196.82 (14.31)	187.11(13.26)	186.65 (22.69)	**0.02 ***	a > b, a > c
Sex, female (%)	70.70	86.36	70.17	55.00	0.08	
Ethnicity (%)						
Italian	85.86	90.09	82.46	90.00		
Other ^2^	14.14	9.10	17.54	10.00		
Adopted (%)	10.10	18.18	10.53	0.00	0.15	
Separated- divorced family (%)	48.48	45.45	54.39	35.00		
Living in a therapeutic community (%)	5.05	9.09	5.26	0.00		
Severe family social difficulties ^3^ (%)	13.13	31.82	8.77	5.00	**0.008 ****	a > b, a > c
Family history of DSM-5 psychiatric disorders (%)						
None	41.41	36.36	49.12	25.00	0.31	
Psychosis	5.05	9.09	3.51	5.00	0.60	
Depression	37.37	22.73	36.84	55.00	0.10	
Bipolar disorder	6.06	0.00	7.02	10.00	0.37	
Anxiety	16.16	13.64	15.79	20.00	0.85	
Substance abuse disorders	15.15	36.36	10.53	5.00	**0.006 ****	a > b, a > c
Eating disorders	5.05	0.00	7.01	0.00	0.44	
Disruptive disorders	5.05	4.54	7.02	0.00	0.46	
Obsessive–compulsive disorders	3.03	9.09	1.75	0.00	0.16	
Personality disorders	1.01	0.00	1.75	0.00	0.69	
Neurodevelopmental disorders	0.00	0.00	0.00	0.00	-	

^1^ Significance: * *p* < 0.05; ** *p* < 0.01. ^2^ Asiatic, African, or American ethnicity. ^3^ Severely impaired family situation (e.g., parent living in prison; parent living in a community for severe drug addiction problems) for which a report has been made to the juvenile court.

**Table 2 brainsci-14-01115-t002:** Personal history of DSM-5 psychiatric disorder, baseline exposure to psychiatric treatments, self-administered questionnaires, and functioning results of the total adolescent sample and the three subgroups [author’s own processing].

Characteristics	Total (*n* = 99)	FBPD (a) (*n* = 22)	HPD (b) (*n* = 57)	OTH (c) (*n* = 20)	*p* ^1^	Post Hoc
Personal history of DSM-5 psychiatric disorders						
Anxiety disorders, *n* (%)	17 (17.17)	4 (18.18)	8 (14.03)	5 (25.00)	0.23	
Psychotic disorders, *n* (%)	20 (20.21)	3 (13.64)	10 (17.54)	7 (35.00)	0.12	
Neurodevelopmental disorders, *n* (%)	17 (17.17)	3 (13.63)	8 (14.04)	6 (30.00)	>0.05	
Bipolar disorders, *n* (%)	16 (16.16)	4 (18.18)	11 (19.30)	1 (5.00)	0.38	
Depressive disorders, *n* (%)	45 (45.46)	7 (31.82)	29 (50.88)	9 (45.00)	0.62	
Obsessive-compulsive disorders, *n* (%)	3 (3.03)	0 (0.00)	0 (0.00)	3 (15.00)	**0.02 ***	c > a, c > b
Post-traumatic or dissociative disorders, *n* (%)	0 (0.00)	0 (0.00)	0 (0.00)	0 (0.00)	-	
Somatic symptoms disorders, *n* (%)	5 (5.05)	2 (9.09)	2 (3.51)	1 (5.00)	0.22	
Eating disorders, *n* (%)	8 (8.08)	1 (4.45)	4 (7.02)	3 (15.00)	0.51	
Conduct disorders, *n* (%)	7 (4.04)	1(4.54)	3 (5.26)	0 (0.00)	0.62	
Substance abuse disorders, *n* (%)	4 (4.04)	2 (9.09)	2 (3.51)	0 (0.00)	0.31	
Baseline exposure to psychiatric treatments						
Antipsychotics, *n* (%)	39 (39.39)	11 (50.00)	22 (38.60)	6 (30.00)	0.41	
Antidepressants, *n* (%)	22 (22.22)	4 (18.18)	13 (22.81)	5 (25.00)	0.86	
Benzodiazepines, *n* (%)	27 (27.27)	8 (36.36)	15 (26.32)	4 (20.00)	0.48	
Mood stabilizers, *n* (%)	11 (11.11)	2 (9.02)	8 (14.03)	1 (5.00)	0.51	
Individual Psychotherapy, yes, *n* (%)	55 (55.55)	13 (59.10)	32 (56.14)	10 (50.00)	0.83	
Parents Psychotherapy, yes, *n* (%)	6 (6.06)	1 (4.54)	5 (8.77)	0 (0.00)	0.57	
Family Psychotherapy, yes, *n* (%)	2 (2.02)	1 (4.54)	1 (1.75)	0 (0.00)	0.35	
Self-administered questionnaires						
CBCL Mother TOT, mean (SD)	69.01 (7.92)	72.26 (8.38)	69.27 (6.84)	65.16 (8.56)	**0.02 ***	a > c
CBCL Mother INT, mean (SD)	69.66 (12.17)	71.26 (8.20)	71.50 (7.37)	63.79 (20.54)	0.29	
CBCL Mother EXT, mean (SD)	64.17 (10.75)	70.79 (9.20)	64.70 (8.50)	56.32 (12.29)	**<0.05 ***	a > c, b > c
CBCL Father TOT, mean (SD)	64.29 (9.19)	66.70 (9.66)	63.68 (9.32)	63.79 (8.98)	0.74	
CBCL Father INT, mean (SD)	66.65 (10.23)	67.50 (10.76)	66.14 (9.63)	67.07 (11.68)	0.81	
CBCL Father EXT, mean (SD)	61.13 (10.71)	65.50 (16.79)	60.57 (9.26)	59.14 (7.55)	0.28	
YSR TOT, mean (SD)	67.97 (11.48)	66.40 (14.17)	70.41 (9.74)	63.30 (11.53)	0.08	
YSR INT, mean (SD)	69.98 (12.47)	64.15 (14.33)	73.31 (10.42)	67.30 (13.14)	**0.02 ***	b > a, b > c
YSR EXT, mean (SD)	63.03 (11.53)	68.30 (9.83)	63.51 (11.05)	56.55 (11.73)	**0.008 ****	a > c
Functioning						
IQ mean (SD)	100.95 (16.25)	91.76 (14.75)	104.64 (15.96)	100.45 (17.59)	**0.009 ****	b > a, b > c
Poor social relations or social withdrawal, *n* (%)	63 (63.64)	13 (59.05)	38 (66.66)	10 (50.00)	0.35	
Poor academic performance or withdrawal, *n* (%)	34 (34.34)	9 (40.91)	20 (35.01)	5 (25.00)	0.72	
Risky behaviors, *n* (%)	67 (67.68)	19 (86.36)	41 (71.93)	7 (35.00)	**0.008 ****	a > c
Clinical Global Impression-Severity (CGI-S), mean (SD)	4.59 (1.04)	4.68 (1.04)	4.75 (0.97)	4 (1.07)	0.06	
CGAS, mean (SD)	44.03 (15.02)	37.68 (15.57)	43.79 (14.03)	51.70 (14.34)	**0.018 ***	c > a

^1^ Significance: * *p* < 0.05; ** *p* < 0.01.

**Table 3 brainsci-14-01115-t003:** Significant differences in socio-demographics and self-administered questionnaires in the adolescent sample that completed follow-up [author’s own processing].

	TPD (a) (*n* = 13)	SPD (b) (*n* = 16)	Non-TPD (c) (*n* = 27)	*p* ^1^ TPD vs. SPD	*p* ^1^ TPD vs. Non-TPD	Post Hoc
Socio-demographics						
Age at baseline, mean (SD)	183 (8.95)	195 (15.5)	190 (14.5)	**0.025 ***	**0.036 ***	**b > a, c > a**
Self-administered questionnaires						
CBCL Mother INT, mean (SD) Missing, *n* (%)	76.0 (6.12) 2 (15.4)	69.8 (9.04) 2 (12.5)	69.9 (6.71) 6 (22.2)	**0.029 ***	**0.039 ***	**a > b, a > c**
YSR INT, mean (SD) Missing, *n* (%)	73.0 (10.2) 1 (7.7)	61.3 (15.0) 1 (6.3)	72.4 (10.0) 3 (11.1)	**0.040 ***	0.842	**a > b**

^1^ Significance: * *p* < 0.05.

## Data Availability

Data are available upon request in Zenodo https://doi.org/10.5281/zenodo.10091173.

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
