# Peer review of "Breakdown or Personality Disorder? Psychiatric Characterization and Developmental Pathways Towards Young Adulthood in Adolescents with Pathological Personality Structure"

_brainsci, 2024, doi:10.3390/brainsci14111115_

Round 1
Reviewer 1 Report
Comments and Suggestions for Authors
Title: Breakdown or Personality Disorder? Psychiatric characterization and developmental pathways towards young adulthood in adolescents with Pathological Personality Structure.
The present manuscript analyses how to address and improve early interventions in young people with a Personality disorder (PD) or in high risk of developing it. From this standpoint, the paper could contribute to literature. Since authors analyze different trajectories, it could be an useful guide to clinicians. However, the manuscript in its present form presents some problems that limit its contribution.
Authors state that clinical structure interviews for DSM5 (SCID-5-PD) and for DSM-IV (SCID-II) were applied. Why both? What did use finally? I understand the former. Am I right?
- It is quite important to state what criteria was used to define the High-risk group.
- Tables 1 and 2 are hard to read. Authors could include significant variables in one table only and report others on supplemental material. It is important to report this information, but it could move to supplemental material. In the same way, paragraphs about table 1 and 2 are also hard to read because is just a list of results with percentages. Authors should try to explain better main results to the readers.
- How do authors define drop-out? Is it just persons that quitted from the study or that present a clinical recovery?
- Table 3 can be deleted.
- In my opinion, the most relevant information of the study is to point out some factors that help us to understand why some people move to a full-blown PD. So, in my opinion is compulsory to add a table with the significant differences between both groups. Note that this table would be the answer to the aim of offering a tentative of why some high-risk adolescents made the transition to full-blown PD.
- Figure 2 can be deleted. This information should be added to the table suggested on the previous point.
- The discussion section is also a list about the role of several factors. Authors should focus on what their results show us about profiling.
- The following sentence has no sense to me: “Finally, there were no statistically significant differences at baseline concerning the main diagnostic instrument, the SCID-5 PD, which is not helpful for clinicians to profile patients at greater risk of transition than others.” How does the main instrument about diagnosing PDs not helpful to clinicians?
- Authors should emphasize the limitation of the low sample size. In fact, considering this low sample size, using terms such as “precise” (line 312) or “substantial” (line 410) are not justify. I understand that the present study is a descriptive one just. No hypothesis or theory is tested. They want to describe tentative differences between some groups. In this way, I think that authors should be more conservative.
- I would appreciate a paragraph on the conclusion summarizing how to profile better an adolescent in high-risk of developing a PD.
As a last point, I have not found the references on the download document.
Minor Points:
- Please, describe in detail what authors consider as a full-blown PD. Is just to present the required number of presented criteria to be diagnosed according to the DSM-5 criteria?
- Why final follow up study population on figure 1 is equal to 101? Do authors also consider excluded people?
- Hypothcontroesize on line 431.
Author Response
Comment 1: Authors state that clinical structure interviews for DSM5 (SCID-5-PD) and for DSM-IV (SCID-II) were applied. Why both? What did use finally? I understand the former. Am I right?
Reply 1: Dear reviewer, thank you for your question. We started our data collection using the DSM-IV and the SCID-II; during our data collection, the DSM-5 and the SCID-5 PD were published in Italian, in 2014 and 2017, respectively. Since our study is observational and naturalistic, we used the most recent version of the DSM and SCID in order to offer our patients the clinical practice as best as possible. To harmonize diagnoses, we updated the DSM-IV ones to the DSM-5 criteria and so we did with the two versions of the SCID. Thanks to your comment, we added specifications in the Materials and Methods section (lines 129-132).
Comment 2: It is quite important to state what criteria was used to define the High-risk group.
Reply 2: Addressing the point you raised, we redefined inclusion criteria for the high-risk group, (lines 151-155).
Comment 3: Tables 1 and 2 are hard to read. Authors could include significant variables in one table only and report others on supplemental material. It is important to report this information, but it could move to supplemental material. In the same way, paragraphs about table 1 and 2 are also hard to read because is just a list of results with percentages. Authors should try to explain better main results to the readers.
Reply 3: We agreed in that we were redundant in reporting data both in the text and the tables. Therefore, we updated the Results section.
Comment 4: How do authors define drop-out? Is it just persons that quitted from the study or that present a clinical recovery?
Reply 4: thank you for your observation: we better clarified the follow-up data collection in the paragraph “2.4.2. Follow-up”: Participants were asked to participate in the follow-up assessment after 1 year from baseline; if they declined to participate, we tried to contact them every year. Those who always declined to participate in the follow-up assessments, were considered as drop-outs. (lines 158-162)
Comment 5: Table 3 can be deleted.
Reply 5: thank you for your suggestion, we considered it as Supplementary material.
Comment 6: In my opinion, the most relevant information of the study is to point out some factors that help us to understand why some people move to a full-blown PD. So, in my opinion is compulsory to add a table with the significant differences between both groups. Note that this table would be the answer to the aim of offering a tentative of why some high-risk adolescents made the transition to full-blown PD.
Reply 6: Thank you for rising this point. We created a table that summarizes our significant results relative to the follow-up analyses, that makes them clearer (Table 3). We also updated the results, reporting (I) significant differences between high-risk patients who made the transition to PD and those who did not (TPD vs non-TPD); (II) significant differences between high-risk patients who made the transition to PD and those who already received a diagnosis (SPD). We updated the text in order to avoid redundancy.
Comment 7:Figure 2 can be deleted. This information should be added to the table suggested on the previous point.
Reply 7: Thank you for your suggestion. Following it, we deleted the figure and added this result in Table 3.
Comment 8: The discussion section is also a list about the role of several factors. Authors should focus on what their results show us about profiling.
Reply 8: Thank you for your comment. Following your suggestion, we updated the discussion section in order to make it more fluent to read.
Comment 9: The following sentence has no sense to me: “Finally, there were no statistically significant differences at baseline concerning the main diagnostic instrument, the SCID-5 PD, which is not helpful for clinicians to profile patients at greater risk of transition than others.” How does the main instrument about diagnosing PDs not helpful to clinicians?
Reply 9: thank you for rising this point. The SCID-5 PD is certainly useful in diagnosing an existent PD and in highlighting the presence of a risk for PDs. On the other hand, our results suggest that the SCID-5 PD is not helpful in discriminating the degree of risk an adolescent patient presents. Indeed, we did not find significant differences in the SCID-5 PD results at baseline between adolescents who later transited to an overt PD and those who did not. We modified the sentence to clarify this issue (lines 370-373).
Comment 10: authors should emphasize the limitation of the low sample size. In fact, considering this low sample size, using terms such as “precise” (line 312) or “substantial” (line 410) are not justify. I understand that the present study is a descriptive one just. No hypothesis or theory is tested. They want to describe tentative differences between some groups. In this way, I think that authors should be more conservative.
Reply 10: thank you for underlying this issue: we used conservative terms and added the small sample size to the limitations of our study (line 395).
Comment 11: I would appreciate a paragraph on the conclusion summarizing how to profile better an adolescent in high-risk of developing a PD.
Reply 11: Thank you for your suggestion. We added a paragraph in the conclusions according to it (lines 375-380).
Comment 12: As a last point, I have not found the references on the download document.
Reply 12: Thank you for rising this point. You are not the only reviewer who reported it, although we included the references in the manuscript we uploaded for revisions. We ensured to include the list of references in the manuscript we modified after revisions, and checked that citations in the text are present in the list of references.
Comment 13: Please, describe in detail what authors consider as a full-blown PD. Is just to present the required number of presented criteria to be diagnosed according to the DSM-5 criteria?
Reply 13: yes, we considered full-blown PDs those who met the DSM-5 criteria. Thanks to your comment, we better defined the inclusion criteria in the FBPD group (line 150). We also added a reference to the DSM-5 criteria in describing the group division of participants at follow-up.
Comment 14: Why final follow up study population on figure 1 is equal to 101? Do authors also consider excluded people?
Reply 14: thank you for highlighting this point, it was a typo and we corrected it in the figure.
Comment 15: Hypothcontroesize on line 431.
Reply 15: thank you for indicating us this typo, we corrected it.

Reviewer 2 Report
Comments and Suggestions for Authors
I thank the authors and the editor for the opportunity to review this interesting manuscript, which describes the results of a small prospective study of personality disorder trajectory among adolescents with a history of hospitalization. The study design and methods are generally adequate and I have only relatively quibbling concerns with the study measures used and with the statistical methods of data analysis which for the purposes of this review are not worth focusing on.
In contrast, I do have several significant concerns with the epidemiological study design, particularly in regards to establishing trajectory. The most significant limitation in this regard is, of course, the very high rate of loss to follow up. Of 99 subjects enrolled, there were only 56 at follow up. Although the authors present helpful data in this regard describing those lost to follow up, the high percentage drop out among their subjects is a significant threat to the internal validity of conclusions regarding trajectory. Likewise, additional secondary limitations, posing similar threats, including the variable duration of follow up, which apparently reflects some degree of right-censoring given the relative recency of the enrollment period through 2021; and a failure to clearly describe
over what period a “loss” is considered to have occurred. The authors should clarify whether (as I assume) a loss is considered any failure to have any period of follow-up. If so, the authors should clarify why they not define a standard follow-up time and assess loss and trajectory at the end of this period, rather than report on trajectory on a maximum follow-up basis as is apparently reported.
It is also not clear from the study methods and Figure 1 whether the n=101 patients assessed for eligibility represent the entire census of patients aged 12-17 admitted during the 2012-2021 time period. This should be stated explicitly and Figure 1 updated as appropriate.
The authors should clarify the above and consider modifications to the methods and discussion in light of these.
Minor Issues:
1. Abstract, line 19. CBCL is not previously defined.
2. Figure 1. “Eligibility”. Remove wavy underscores throughout. Correct final follow up study population (i.e., 99-43 = 56, not 101 as listed).
3. Conclusions. Line 421. Clarify the described enrollment of outpatients. Were not all subjects enrolled as inpatients?
Author Response
Dear reviewer, thank you for your revisions.
Comment 1: I do have several significant concerns with the epidemiological study design, particularly in regards to establishing trajectory. The most significant limitation in this regard is, of course, the very high rate of loss to follow up. Of 99 subjects enrolled, there were only 56 at follow up. Although the authors present helpful data in this regard describing those lost to follow up, the high percentage drop out among their subjects is a significant threat to the internal validity of conclusions regarding trajectory.
Reply 1: Thank you for highlighting this point, after your revision we included it in the limitations of our study (line 396). Since our study is observational and naturalistic, and patients from all Italy access our hospital, it is not infrequent that patients that come from different and far regions are lost in time. Moreover, the focus of our study was on patients who presented a high risk for PDs or an overt PD at baseline, and the rate of drop-out among these patients equally distributed between the two groups (FBPD and HPD). From a statistical point of view, our conclusions are still valid for the populations represented at follow-up, with the intrinsic limitations of a small sample size. Therefore, although we agree that it would have been interesting and more complete to have a higher percentage of follow-up completed and data from the OTH group, we believe that our conclusions are not threated by the high drop-out rate.
Comment 2: Likewise, additional secondary limitations, posing similar threats, including the variable duration of follow up, which apparently reflects some degree of right-censoring given the relative recency of the enrolment period through 2021; and a failure to clearly describe over what period a “loss” is considered to have occurred. The authors should clarify whether (as I assume) a loss is considered any failure to have any period of follow-up. If so, the authors should clarify why they not define a standard follow-up time and assess loss and trajectory at the end of this period, rather than report on trajectory on a maximum follow-up basis as is apparently reported.
Reply 2: Thank you for rising this point. In our study, we contacted patients for follow-up at 1 year from baseline. If they declined to participate to the follow-up assessment, we tried to contact them every year. We considered those who never agreed to participate in the follow-up assessments as participants who dropped out. Participants could have accepted to be interviewed at any time point after baseline, and this is the origin of the high variability in the follow-up intervals. In this regard, we updated the paragraph “2.4.2. Follow-up” clarifying it (lines 158-162).
Comment 3: It is also not clear from the study methods and Figure 1 whether the n=101 patients assessed for eligibility represent the entire census of patients aged 12-17 admitted during the 2012-2021 time period. This should be stated explicitly and Figure 1 updated as appropriate.
Reply 3: Thank you for your suggestion, we updated our figure according to it.
Comment 4: Abstract, line 19. CBCL is not previously defined.
Reply 4: Following your comment, we updated the abstract.
Comment 5: Figure 1. “Eligibility”. Remove wavy underscores throughout. Correct final follow up study population (i.e., 99-43 = 56, not 101 as listed)
Reply 5: Thank you for lighting this typo up, we corrected it.
Comment 6: Conclusions. Line 421. Clarify the described enrolment of outpatients. Were not all subjects enrolled as inpatients?
Reply 6: Thank you for your comment. We enrolled both inpatients and outpatients that accessed our Child and Adolescent Neuropsychiatry Unit; outpatients accessed the facility on a weekly basis to complete psychiatric assessment. Following your comment, we updated the description in paragraph 2.3. Study population (lines 96-97)
Reviewer 3 Report
Comments and Suggestions for Authors
well done!

Author Response
Dear reviewer, thank you for your kind revisions.
Comment 1: From line 66 to 76 the text seems redundant, the aim has been well described in the previous paragraph.
Reply 1: Thank you for your observation. We agreed with you and deleted the last part of the introduction.
Comment 2: The paper does not have a list of bibliographic citations; references from the literature may be the real added value of the entire work.
Reply 2: Thank you for mentioning this point. Although we included the list of references in the manuscript we uploaded, you are not the only reviewer to have pointed this out. We made sure to include the list of references in the manuscript we edited after revisions.
Round 2
Reviewer 1 Report
Comments and Suggestions for Authors
I think that authors have made a great effort to improve the paper. All points have been properly addressed. Now, the main concerns about the paper are mainly reduced. I have only comment. I realize that some post-hoc comparisons are strange. For instance, they state on table 1 that group a score higher than b on “Severe family social difficulties”, but the group c presents the lowest mean. A similar pattern is reported for substance abuse results. On table 2, they report b>c on “YSR INT, mean” but the lowest mean is for group a. Again, a similar pattern is reported for IQ mean. I have found another strange results on the post-hoc analyses. I think that authors should review the groups where significant differences are reported on the post-hoc analyses of all tables.
Author Response
Dear reviewer,
Thank you for your comments that helped us improving our paper.
Comment 1: I think that authors should review the groups where significant differences are reported on the post-hoc analyses of all tables.
Reply 1: Thank you for rising this point. We reviewed and updated the tables according to your suggestion.